# Perovskites fabricated on textured silicon surfaces for tandem solar cells

Sang-Won Lee[1], Soohyun Bae[1], Jae-Keun Hwang[1], Wonkyu Lee[1], Solhee Lee[1], Ji Yeon Hyun[1], Kyungjin Cho[1], Seongtak Kim[2], Friedemann D. Heinz[3], Sung Bin Choi[4], Dongjin Choi[1], Dongkyun Kang[1], Jeewoong Yang[1], Sujeong Jeong[1], Se Jin Park[1], Martin C. Schubert [5], Stefan Glunz[5,6], Won Mok Kim[4], Yoonmook Kang[7], Hae-Seok Lee[7 ✉] & Donghwan Kim[1,7 ✉]

The silicon surface texture significantly affects the current density and efficiency of perovskite/silicon tandem solar cells. However, only a few studies have explored fabricating perovskite on textured silicon and the effect of texture on perovskite films because of the limitations of solution processes. Here we produce conformal perovskite on textured silicon with a dry two-step conversion process that incorporates lead oxide sputtering and direct contact with methyl ammonium iodide. To separately analyze the influence of each texture structure on perovskite films, patterned texture, high-resolution photoluminescence (μ-PL), and light beam-induced current (μ-LBIC), 3D mapping is used. This work elucidates conformal perovskite on textured surfaces and shows the effects of textured silicon on the perovskite layers with high-resolution 3D mapping. This approach can potentially be applied to any type of layer on any type of substrate.

---

[1] Department of Materials Science and Engineering, Korea University, Seoul 136-713, Republic of Korea. [2] Gangwon Regional Division, Korea Institute of Industrial Technology, Gangwon-Do 210-340, Republic of Korea. [3] Laboratory for Photovoltaic Energy Conversion, Department of Sustainable Systems Engineering (INATECH), University Freiburg, Emmy-Noether Strasse 2, 79110 Freiburg, Germany. [4] Korea Institute of Science and Technology (KIST), Seoul 02792, Republic of Korea. [5] Fraunhofer Institute for Solar Energy Systems ISE, Freiburg 79110, Germany. [6] Laboratory for Photovoltaic Energy Conversion, University Freiburg, Freiburg 79110, Germany. [7] KU·KIST Green School, Graduate School of Energy and Environment, Korea University, Seoul 136-713, Republic of Korea. ✉email: lhseok@korea.ac.kr; solar@korea.ac.kr

The power conversion efficiency (PCE) of perovskite solar cells has significantly increased from 3.81 to 25.2% (refs. [1–3]) in the past nine years. In the case of silicon solar cells, a record efficiency of 26.7% has been reported, which is very close to the theoretical efficiency limit[1,4–6]. To overcome the efficiency limit of a single-junction device, a perovskite/silicon tandem approach can be used because of several advantages of perovskite solar cells, including tunable bandgap[7,8], easy fabrication[9,10], and high efficiency[11,12]. To produce perovskite/silicon two-terminal tandem solar cells, several hundred nanometers of conformal perovskite layer should be fabricated on a micrometer-sized pyramidal textured silicon surface because high-efficiency silicon solar cells require surface texture to maximize light absorption[6,13–15]. However, most perovskite solar cells are manufactured by solution-based processes that cannot be used on the micrometer-sized textured structure. Not only perovskite absorber layer, but also the other components like electron transfer layer (ETL) and hole transfer layer (HTL) have been mainly produced by solution process. As a consequence, most two-terminal perovskite/silicon tandem solar cells have been produced with flat silicon front surfaces[16–23]; flat surfaces restrict the light absorption and current density, which has the greatest effect on the PCE, as shown in Fig. 1. Figure 1 was constructed based on the literature results as shown in Supplementary Table 1. The red lines and numbers denote the results of a linear fitting and the slope, respectively. The slopes had been obtained by linear regression fitting. Statistically, the current density exhibits the greatest correlation with the PCE. If we can obtain 1 mA/cm$^2$ more current density, 1.61 percentage point more efficiency will be obtained. The use of antireflection foils on top of the tandem device has been studied as one way to achieve high current density. Referring to literature results,

however, eventually forming perovskite on textured silicon will generate maximum current density[24–26]. Therefore, a technique to produce a conformal perovskite layer and other solar cell components on a textured surface is required. Perovskite solar cell fabrication processes can be classified into five methods (Supplementary Tables 1 and 2). One of the strong candidates is the evaporation incorporated process[27–29]. With a hybrid process combining evaporated precursor and solution-based conversion processes, Sahli et al. reported a 25.2% perovskite/fully textured silicon tandem solar cell[30,31]. However, the processing conditions for organic material evaporation are challenging, and problems can exist in the large area process. Another competitive candidate is the dry two-step conversion process[32–35]. If the conformal precursor can be secured, a conformal layer and large area uniformity can be expected. Superior process flexibility is also obtainable by controlling the conversion conditions and precursor properties, such as composition, morphology, and crystallinity, which can be managed by pretreatment or posttreatment.

Here, we propose a way to produce conformal perovskite on textured silicon with a dry two-step conversion process, and investigation with patterned texture and high-resolution 3D mappings.

## Results and discussion

**Perovskites on flat surfaces.** We first tried to produce a conformal perovskite layer by a one-step spin-coating method and a two-step hybrid method. We had adopted the PbO precursor that can be easily deposited by sputtering process. Perovskite solar cells with PbO-based conversion process previously reported on flat surface[36–38]. Zhang Z. et al. had reported 14.1% with

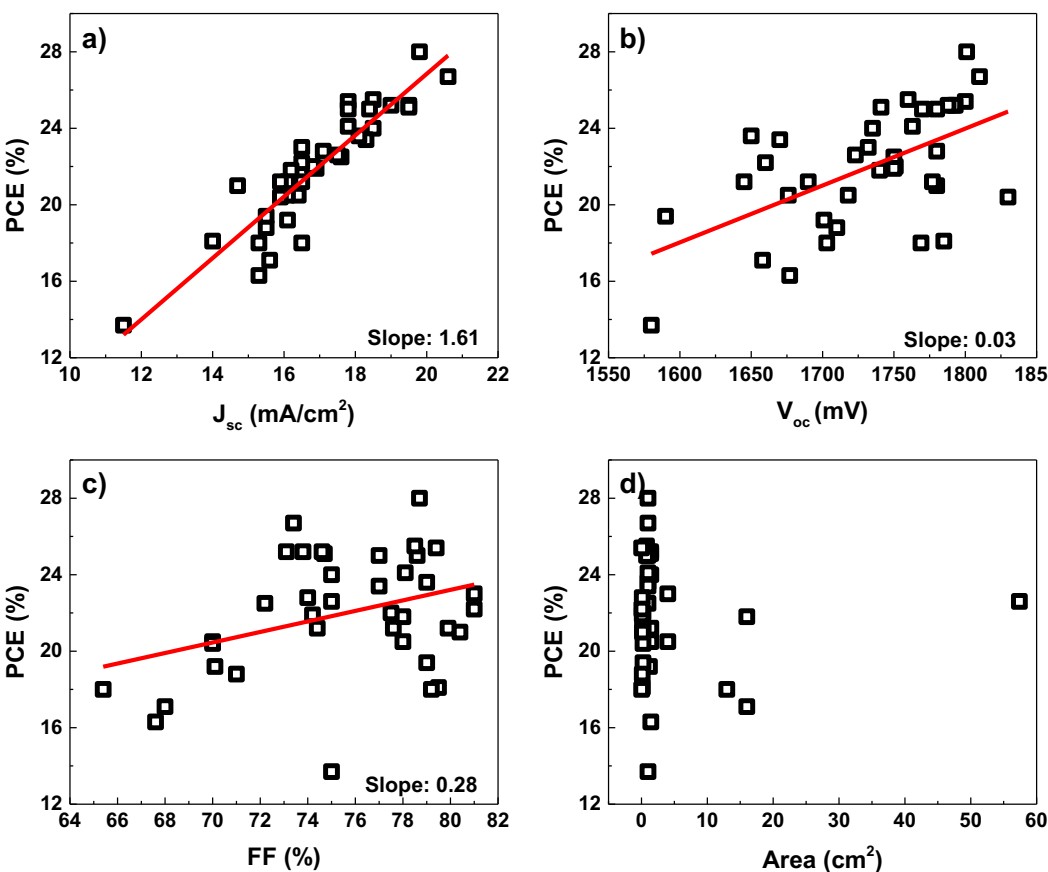

**Fig. 1 Relationship between efficiency and parameters of perovskite/silicon two-terminal tandem solar cells.** Dependence of efficiency on **a** short-circuit current density, **b** open-circuit voltage, **c** fill factor, and **d** device area.

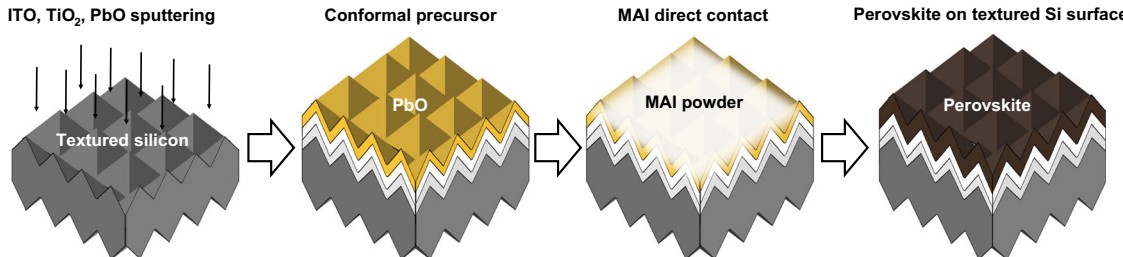

**Fig. 2 Schematic of a proposed dry two-step conversion process for conformal perovskite fabrication on textured silicon surfaces.** Indium tin oxide (ITO), titanium dioxide (TiO$_2$), and PbO layers were first deposited on a textured silicon surface by means of sputtering to form the bottom electrode, ETL, and precursor, respectively. Subsequently, a perovskite layer was converted by direct contact with MAI powder and annealing.

sputtered PbO and MAI solution dipping conversion process[36]. Yan-Lin Song et al. had reported 14.6% with PbO electro deposition and MAI spin-coating conversion[37,39]. In our case, with the textured surface and a solution-based process, uniform films were not obtained even with a conformal precursor (Supplementary Fig. 1). As a next step, the dry two-step conversion process was investigated and Fig. 2 illustrates the proposed process. The PbO precursor was first deposited by sputtering process, and converted into perovskite by direct contact reaction with MAI and annealing.

The dry two-step conversion was first verified with flat fluorine-doped tin oxide (FTO) substrates. Conformal 80 nm PbO films were sputtered and converted into CH$_3$NH$_3$PbI$_3$ layers through a direct contact reaction with CH$_3$NH$_3$I powder at 100, 150, 170, 200, and 250 °C for 35, 70, 140, and 210 min. As shown in Fig. 3a, as reaction temperature and time increased, the films gradually changed to yellow PbI$_2$ and then turned into a dark brown colored perovskite. In the case of 250 °C, MAI was burned, and in the process of removing reminded MAI by nitrogen gas (N$_2$) blowing and isopropyl alcohol (IPA) rinsing after the reaction finished, all the films were removed together. These tendency was also observed at light absorptance, scanning electron microscopy (SEM) image, and normalized X-ray diffraction (XRD) results as shown in Fig. 3b–d. Normalization was conducted based on the maximum intensity to the ratio of PbO$_x$, PbI$_2$, and CH$_3$NH$_3$PbI$_3$ intensity. The peak positions of each component were 30.5°, 12.8°, and 14.3° 2-theta degrees[37]. Supplementary Figs. 2–7 show additioal informations, including the light absorption at the full wavelength, the cross section of the SEM, the diffraction intensity of the XRD at the entire angle, shorter time conversion result at 200 °C, 250 °C, tauc plot result for optical bandgap calculation, and the effect of the MAI amount on the conversion process. The dependence on the amount of MAI in the reaction was not significant within the experiment conditions.

To the next, perovskite solar cells were produced based on the investigated conversion conditions. Figure 4 shows the solar cell parameter distributions produced at different conversion condition. Amount of MAI was fixed at 5.0 g. A 11.1% PCE was obtained at 200 °C 70 min conversion condition. For the perovskite solar cell, we used TiO$_2$ ETL and 2,2′,7,7′-tetrakis(N,N-di-p-methoxyphenyl-amine)-9,9′-spirobifluorene (Spiro-MeO-TAD) HTL, which achieved ~20% PCE in our previous paper[40]. The LIV curves and the current tracking results for each solar cells are shown in the Supplementary Figs. 8 and 9, and Supplementary Tables 3 and 4.

**Perovskites on textured silicon surfaces**. A conformal precursor is a prerequisite for conformal perovskite on a textured silicon surface. The 4-inch uniformity of the sputtered PbO precursor films on the textured surface was demonstrated by the thickness, crystallinity, and chemical bonding ratio, SEM, XRD,

and X-ray photoelectron spectroscopy. A thickness uniformity of 8.3 and 4.0% chemical bonding ratio uniformity were obtained (Supplementary Figs. 10 and 11, Supplementary Table 5). With this uniform precursor layer, the dry two-step conversion process was performed. The obtained perovskite film quality was investigated by SEM and high-resolution photo-luminescence (μ-PL)[41,42] 3D mapping, which will be described in the next section. A conformally deposited perovskite film was observed, as shown in Fig. 5a–f. Less than 10% optical reflectance was obtained in the range 300–1100 nm, as shown in Fig. 5g; this effect can contribute to high current density output for tandem solar cells. Figure 5h, i shows the μ-PL emission peak position and peak intensity across the scanning area of ~20 μm × 20 μm. An emission peak position of 765–770 nm, which is well matched with perovskite bandgap energy and the other group reports was observed[41]. In the case of PL intensity, there was very large variation from 0 to 1. This variation in PL intensity could be an indication of the perovskite quality difference depending on the film location above the substrate texture, for example, on a pyramid tip or in a valley.

**Detailed analysis with V-groove texture and 3D mapping**. Few studies have explored the influence of textured silicon on perovskite films because of several obstacles in investigation. (1) Producing perovskite on textured silicon surfaces has limitation in fabrication process. (2) The random nature of conventional textured silicon makes it difficult to find exact relations between the texture structure and perovskite film. (3) The thickness scale difference between the textured silicon and the perovskite layer makes it difficult to choose an analysis tool. Therefore, an appropriate fabrication process and analysis method are required. The production of a perovskite layer on a textured surface is solved by the dry two-step conversion process. To settle the problem of randomness, we used periodically structured textures, namely, V-groove textures[15] with flat surfaces and tips, as shown in Fig. 6. μ-PL and μ-light beam-induced current (μ-LBIC)[41,42] 3D mapping, which can show not only comprehensive but also local information about films and devices, were adopted as analysis methods. Since several hundred nanometers of perovskite layer is located on the micrometer-sized pyramidal textured silicon surface, and the measurements have to maintain high spatial and depth resolution simultaneously, only XY 2D mapping cannot provide accurate information, as shown Supplementary Fig. 14. To overcome these issues, we applied XYZ 3D mapping. Several XY 2D maps were measured first for different depths Z, as shown in Fig. 7a, b. The strongest PL signal among the different Z was adopted as representative PL data at that XY position, and this depth Z was used to construct the focus height. All the collected images were combined into a final 2D image. Figure 7c–e represents combined 2D images of the focus height, normalized PL intensity, and peak position, respectively. Figure 7f shows the

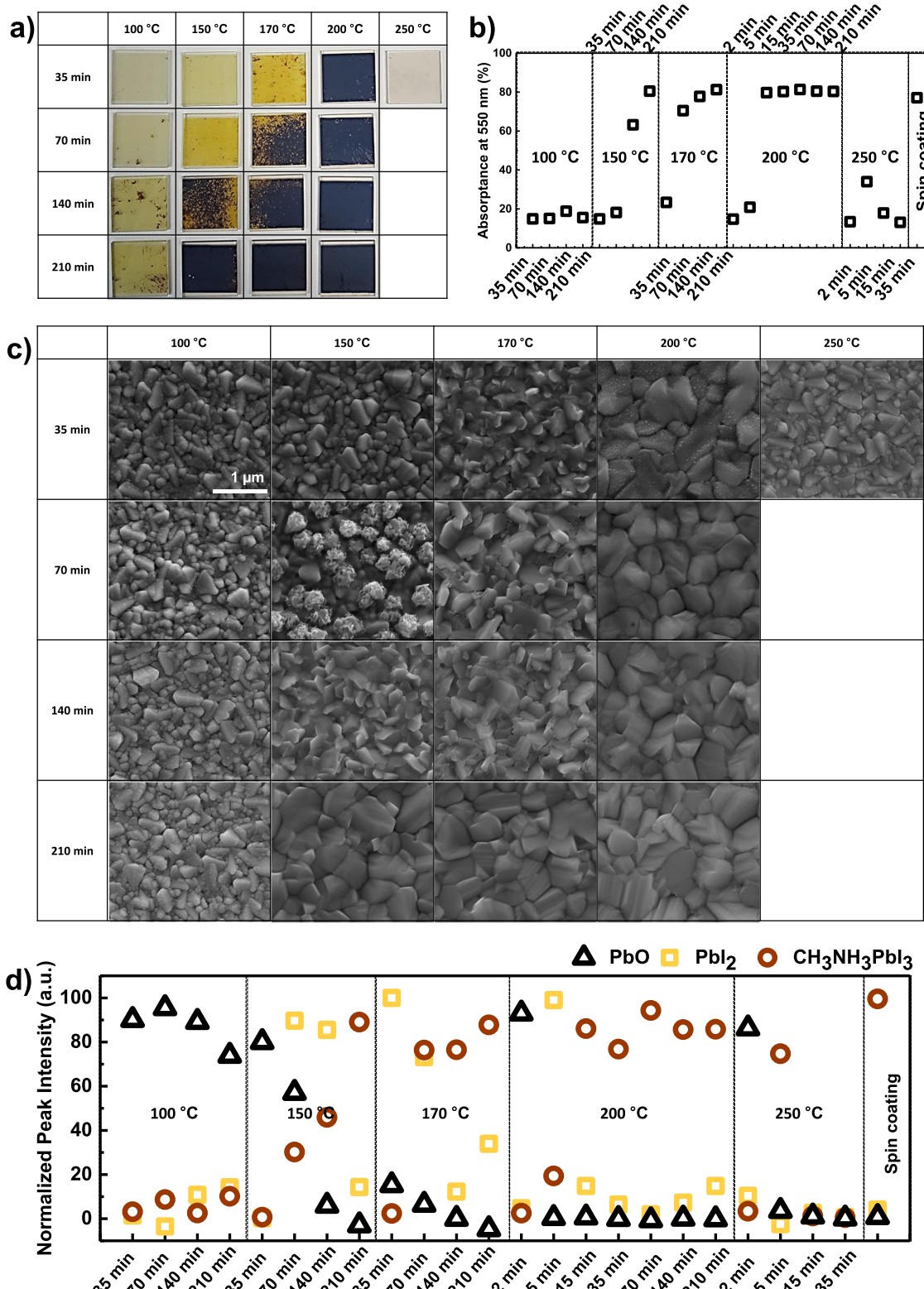

**Fig. 3 Characteristic of films produced by the dry two-step conversion on a flat surface with different conversion time and temperature. a** Digital camera image of corresponding films on a 1.5 cm × 1.5 cm TiO$_2$/FTO substrate, **b** light absorptance at 550 nm, and **c** SEM top view. **d** Normalized XRD peak intensity at 30.5° for PbO, 12.8° for PbI$_2$, and 14.3° for CH$_3$NH$_3$PbI$_3$.

SEM top view of the corresponding device. The focus height well follows the adopted V-groove texture topography, and this finding means the measurement was conducted with in focus. PL emission was obtained at ~764–768 nm.

In the case of PL intensity, there was a very large distribution from 0 to 1, similar to the result shown in Fig. 5i. With the patterned texture, we could clearly designate the PL peak position and PL intensity variation to a specific location of the textured

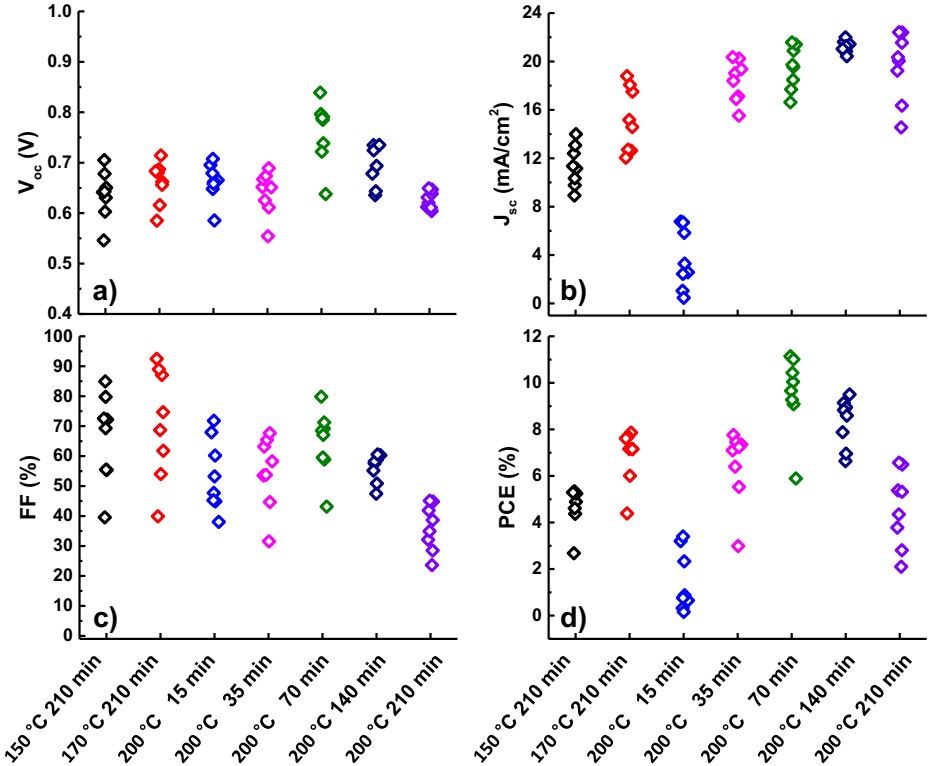

**Fig. 4 Solar cell parameter distributions converted at different conditions.** Eight solar cells were produced for each condition. **a** Open-circuit voltage, **b** short-circuit current density, **c** fill factor, and **d** power conversion efficiency. Hysteresis was large when measuring solar cell efficiency, and there were cases where FF was over estimated. Each data point was adopted from the LIV reverse scan results and the measurements were conducted with 0.078 cm$^2$ shadow mask under the AM1.5 G condition. The area of the mask was measured by optical microscope and equipment at Korea Institute of Energy Research.

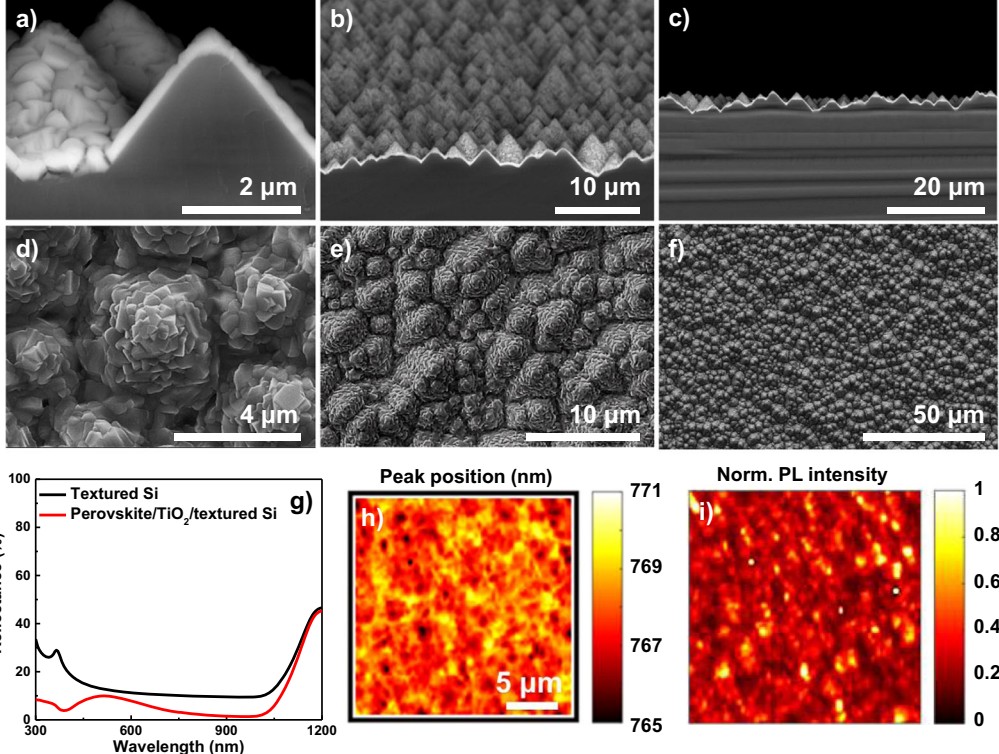

**Fig. 5 Conformal perovskite on a randomly textured silicon surface. a–c** SEM image of conformal perovskite on silicon with backscattered electron mode; **d–f** top view SEM image with secondary electron mode; **g** reflectance of perovskite produced using the dry two-step process on randomly textured silicon; **h** μ-PL peak position; and **i** normalized PL emission intensity. The sample structure is perovskite/TiO$_2$/silicon.

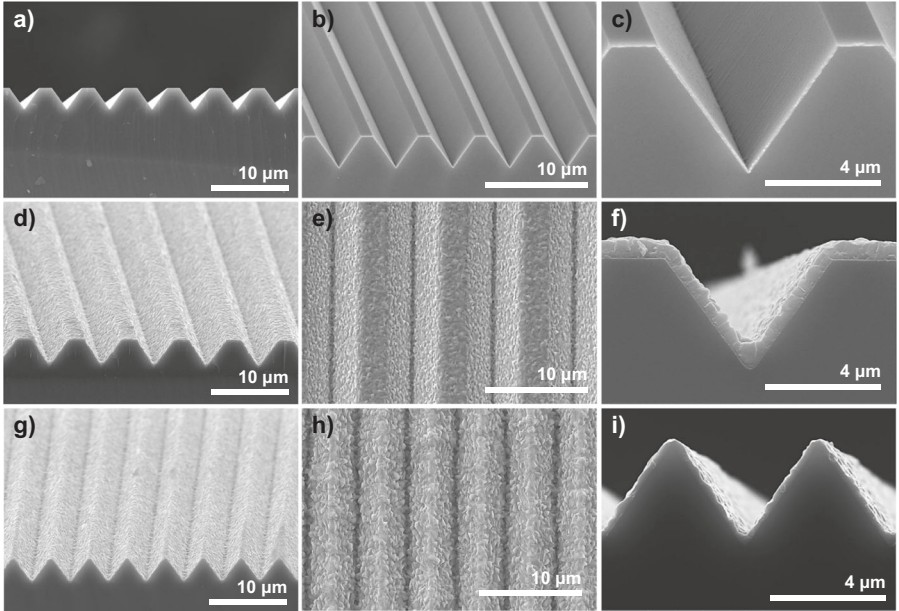

**Fig. 6 SEM image of a periodically patterned texture and layer of perovskite fabricated on it. a** V-groove textured substrate produced by photolithography and wet etching. **b**, **c** TiO$_2$ and PbO deposited on the V-groove textured silicon. **d–f** SEM images of perovskite on the V-groove texture with flat surfaces, and **g–i** tips. The interval and size of the patterned texture were controlled based on the experimental purpose. The detailed process is described in the Methods section. The optical reflectance data for the V-groove texture are also provided in Supplementary Fig. 12 and Supplementary Table 6.

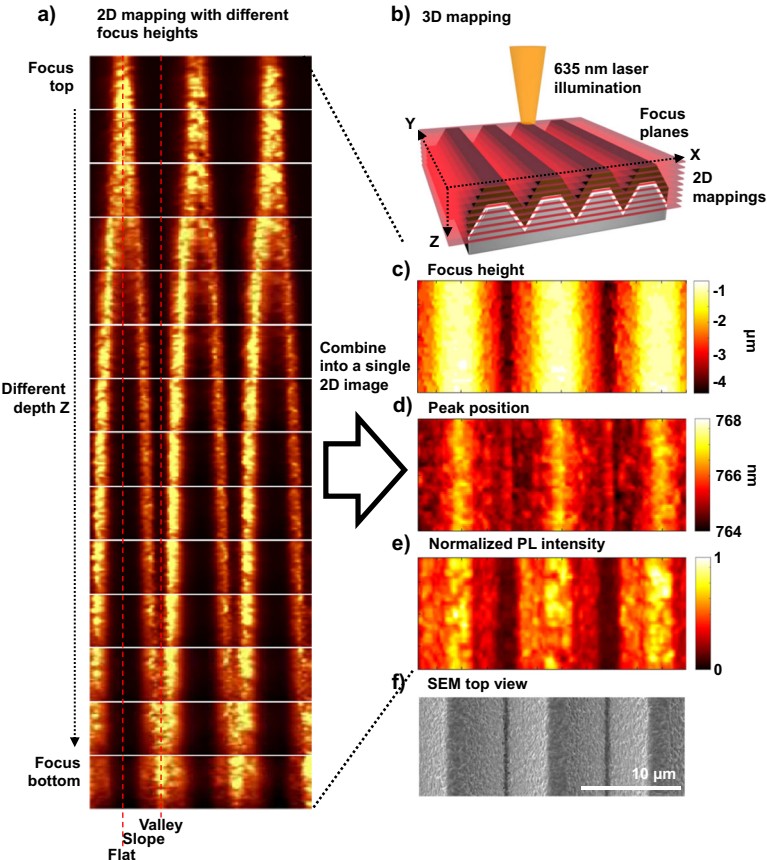

**Fig. 7 Results of 3D mapping analysis for perovskite on periodically textured silicon. a** μ-PL 2D maps with different depths. **b** Schematic diagram of the μ-PL and μ-LBIC 3D mapping method. **c–e** Combined image of the collective 2D maps. **c** Focus height, **d** PL peak position, **e** normalized PL intensity, and **f** corresponding SEM image. The results of PL mapping using different size V-groove textures are in Supplementary Fig. 13. The PL intensity and the peak position dependency on location were the same even when the length of the slope was changed. Displaying the PL spectrum at a specific position for specific focus heights are shown in the Supplementary Fig. 14. Measurements were conducted with a spatial raster of 250 nm and Z distance of 250 nm. The details of the measurement setup are described in the characterization section.

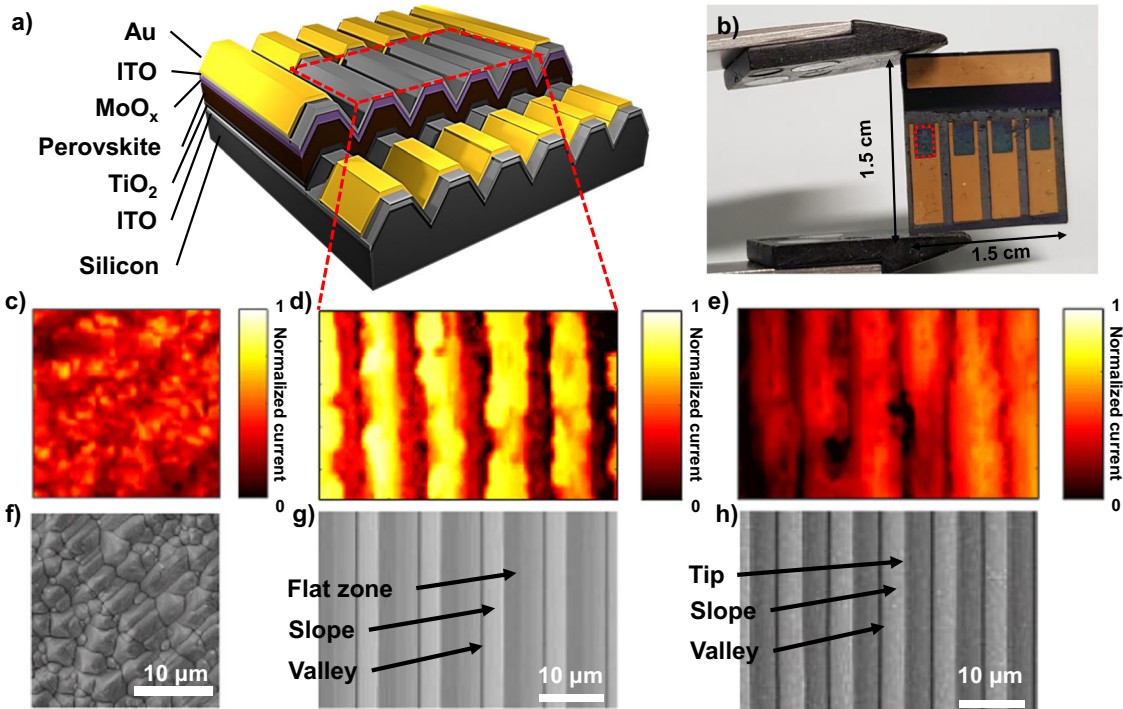

**Fig. 8 Result of perovskite solar cell fabrication and analysis on patterned silicon texture. a** Schematic diagram of a perovskite solar cells on a textured silicon surfaces. **b** Digital camera image of an actual device. The inset red dotted line square in **a** corresponds to the inset red dotted line square in **b** and measurement result in **d**. µ-LBIC 3D mapping results of the perovskite solar cell on **c** random texture, **d** V-groove with flat zones, and **e** V-groove with tips. Most of the dark areas of the PL and LBIC mapping actually showed very small values rather than actual zero. But there were actually zero part in the raw data. These parts can be shunt points. This kind of problem can be occurred during the deposition of HTL, buffer layers, and transparent electrodes sputtering process. **f–h** show SEM images of each device. The SEM images are obtained from different areas of the corresponding samples.

silicon. This result is evidence of the film quality variation affected by particular substrate structures and locations. Longer wavelengths and higher intensities were observed at flat surfaces. Shorter wavelengths and lower intensities were observed at the edges of flat surfaces and at valleys, where the substrate was bent. To understand the relationship between PL difference and device performance, we applied µ-LBIC 3D mapping to a completed device with a structure of Au/ITO/MoO$_x$/perovskite/TiO$_2$/ITO/ textured Si, as shown in Fig. 8. The ITO electrodes and ETL were fabricated using a sputtering process. MoOx (refs. [43–45]; which was adopted as HTL) and the Au electrode were produced through thermal evaporation. In the case of a device on a conventional texture, it is difficult to accurately determine the relationship between the amount of current and the substrate structure. However, certain current patterns were identified when V-groove textures were applied. The highest current outputs were obtained at the flat surfaces, and relatively low current outputs were observed at the tips and valleys of the V-groove texture. These results indicate an adverse effect of the substrate on the perovskite device at the locations where the substrate is bent. These locations correspond to where the shorter PL emission wavelengths and lower intensities were obtained. Similar results were obtained with c-AFM leakage current mapping. Supplementary Fig. 15 shows the 3D and 2D views of the leakage current mapping results. The leakage current occurred periodically at the edges of the flat surfaces. The low performance at bent locations can be attributed to the stress induced on the perovskite by the textured silicon during the conversion process.

**Stress analysis and methods for stress relaxation.** The stress affects the electronic structure, carrier mobility, and corresponding perovskite solar cell performance[46–50]. Approximately

two-fold volume expansion was reported when using PbI$_2$ precursor[51], and we observed approximately five-fold volume expansion during the PbO conversion process (Supplementary Fig. 16). To investigate stress inducement, Silvaco Athena (ver.5.22.1.R) software simulation, which is specialized in stress simulation was performed using the perovskite and silicon material constants[52,53]. The detailed equations and results are shown in Fig. 9 (Supplementary Table 7, Supplementary Equation 2). The perovskite materials on the V-groove texture was constructed. A temperature of 200 °C was applied, and then, the material was cooled down to room temperature. To express the 3D stress in 2D, stresses applied in the $x$-direction of the $x$-plane (Fig. 9a, d, e) and the $y$-direction of the $y$-plane (Fig. 9b, f, g) were calculated, and expressed in the $z$-plane (Fig. 9c). The most stressed area corresponded to the valley and perovskite/Si interface near the valley, as shown in Fig. 9e, g, followed by the slope and edges of the flat surfaces.

Given the relationships among the stress simulation, µ-PL and µ-LBIC 3D mapping, and the c-AMF measurements, the locations where lower device performance was obtained were the locations where the stress applied to the device changed significantly. This occurred where the structure of the substrate was bent. As a possible solution to reduce the stress of perovskite on textured surfaces, we investigated using porous precursor and substrate chemical rounding methods[54]. These processes can produce spare space for the conversion process, and soften the bent and sharp structures. The porous precursor was fabricated by treating PbO with HF vapor, which was generated at 40 °C. Substrate rounding was performed by dipping the textured silicon substrate in a 1:50 (% (v/v)) solution of diluted HF and HNO$_3$ at room temperature[54]. Supplementary Figs. 17 and 18 show the results of HF treatment and chemical rounding. With increasing HF exposure time, larger

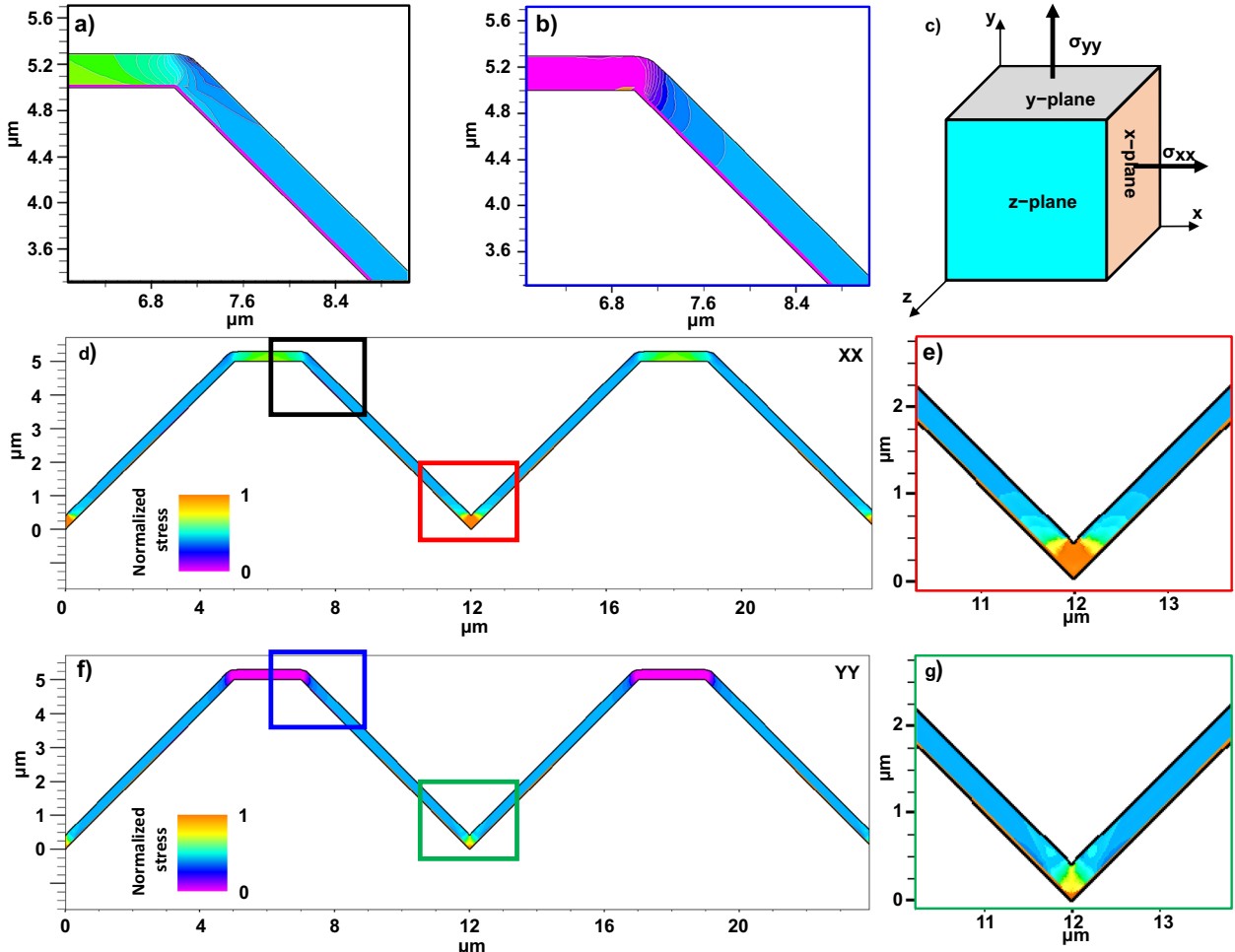

**Fig. 9 Stress simulation results of perovskite solar cells on a patterned silicon texture. a**, **d**, **e** Stresses applied in the *x*-direction of the *x*-plane and **b**, **f**, **g** the *y*-direction of the *y*-plane are expressed in 2D at the *z*-plane. **a** and **e** present the magnified views of the inset squares in **d**. **b** and **g** present the magnified views of the inset squares in **f**. The amount of stress varies based on how the initial internal stress is set, although the tendency of the stress distribution does not change.

perovskite grains were observed. Increasing rounding times gave more spaces for perovskite conversion, particularly at the valley as shown in Supplementary Figs. 17–19. Supplementary Fig. 19 shows a normalized XRD peak comparison at the (110) and (310) planes (Supplementary Figs. 20 and 21, Supplementary Table 8). Perovskite produced by the one-step spin-coating method on a flat surface shows XRD peaks at 14.05° and 31.81°. However, for the dry two-step conversion process on the textured surface, the XRD peaks shifted to 14.20° and 31.99°. By applying the HF-treated precursor, the peaks shifted to 14.16° and 31.91°, which are closer to those of the reference film. In contrast, the substrate rounding process was not effective in reducing the peak shift. Based on the XRD results and previously reported elastic constants of $CH_3NH_3PbI_3$ (refs. [55,56]), the amount of stress was calculated. If we assume that the one-step spin-coated perovskite is stress free, ~106–212 MPa stress is induced in the perovskite film on the textured surface. With the HF-treated precursor, the stress is reduced to ~77–155 MPa. The effect of HF treatment was also investigated by μ-PL 3D mapping, as shown in Fig. 10. HF treatment reduced PL intensity and emission peak differences among the entire area and surfaces. This finding possibly means that this treatment reduced the stressed area and adverse effects of substrates. As a consequence, porous precursors have the potential to be applied in producing high-efficiency perovskites on textured silicon surfaces in extended studies.

In conclusion, we tried to address the problems of (1) producing conformal perovskite on textured silicon surfaces, (2) analyzing perovskite on textured surfaces caused by the randomness of conventional textured silicon, and (3) the size-scale difference between the textured silicon and the perovskite layer. We demonstrated the dry two-step conversion process to produce conformal perovskite solar cells on a textured silicon surface. This process can also be applied to upscaling of perovskite as shown in our previous result[57]. To clearly distinguish the effects of different texture structures, we adopted a patterned texture, the so-called V-groove texture. The produced perovskite layers were investigated in detail with μ-PL and μ-LBIC 3D mapping, stress simulation, and c-AFM. We observed a disadvantageous effect of substrates on the device performance at the bent and stressed positions. As a way to enhance perovskite film quality, we proposed HF treatment and substrate chemical rounding processes, and HF treatment showed an ~50 MPa stress-relaxing effect and enhanced film quality.

The dry two-step conversion process investigated here can regulate the composition and quality of the perovskite film by controlling the conversion process or precursor compositions. Posttreatments and pretreatments are also productively adoptable, as shown in this paper. The fabrication and analysis method introduced here can be applied to any type of layer on any type of substrate.

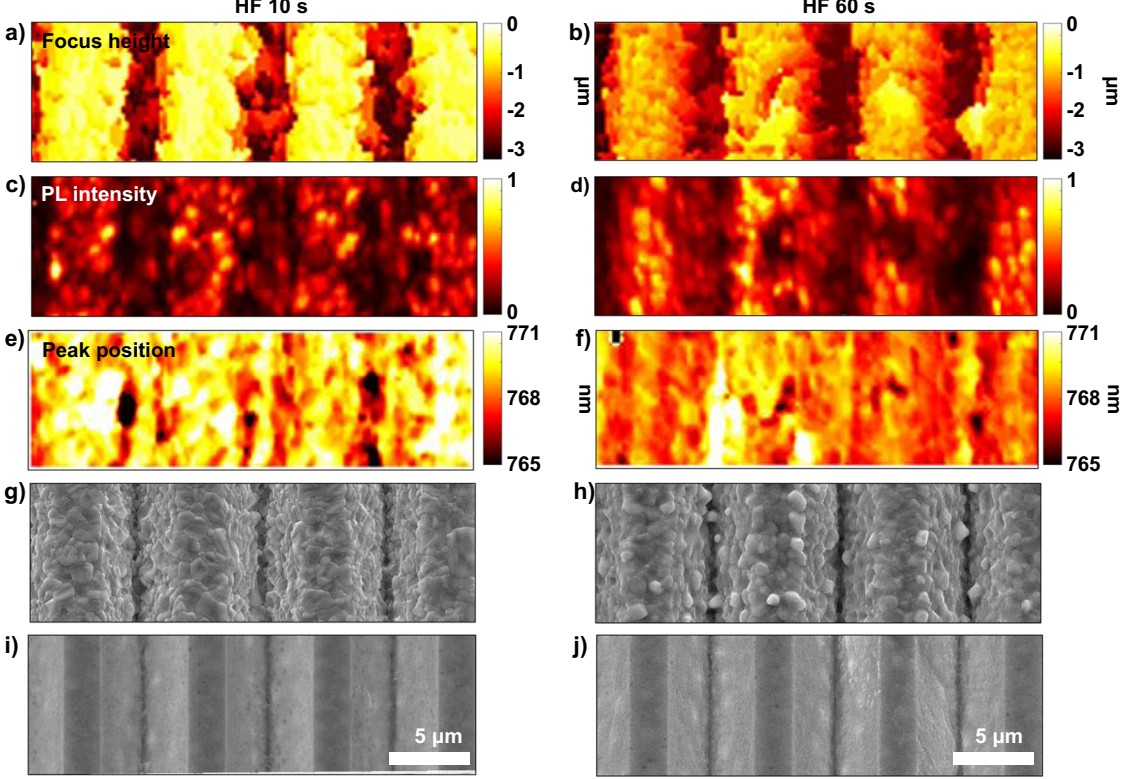

**Fig. 10 μ-PL 3D mapping results of perovskite fabricated with HF-treated PbO precursors. a, b** Focus height, **c, d** normalized PL intensity, **e, f** peak position, **g, h** corresponding SEM images of converted perovskite, and **i, j** HF-treated PbO precursor.

## Methods

**Conformal perovskite layer and solar cell fabrication.** FTO glass substrates (7 Ω/sq), 600 μm n-type polished semiconductor wafers, and silicon solar wafers, which were textured on both sides, were used as substrates. The flat substrates were cleaned with acetone, ethanol, and IPA prior to being used for producing perovskite solar cells. Subsequently, an ultraviolet ozone treatment was applied for 30 min. To create a device on the textured silicon surface, a transparent bottom ITO electrode was produced by sputtering over the differently textured substrates. With respect to the conformal ETM, $TiO_2$ was fabricated by RF magnetron sputtering. A conformal $PbO_x$ precursor layer was also deposited by RF magnetron sputtering. The precursor films were converted into perovskite by direct contact with MAI powder (Greatcell Solar). The MAI powder was directly spread over the precursor film and annealed at respective temperature for respective times in air. After the reaction finished, remaining MAI were removed by $N_2$ gas blowing, IPA rinsing, and dried by spinning. Similar to conformal HTL, $MoO_x$ was fabricated by thermal evaporation. Subsequently, a 150 nm ITO layer was formed through sputtering, and a 100 nm Au electrode was finally deposited by thermal evaporation.

**V-groove fabrication.** V-groove substrates were fabricated by combining photo-lithography techniques with wet etching[15]. First, 200 nm SiNx etching barriers were deposited by plasma-enhanced chemical vapor deposition on a 600 μm n-type polished semiconductor wafer. To form a specific pattern on the wafer, a 2 μm coating of positive photoresist (PR) was formed by spin coating and was exposed to light for 2.8 s with a pattern mask. Subsequently, pattern development was performed. The PR deposition and pattern-developing processes were conducted at the Korea Advanced Nano Fab Center (KANC). The etching barriers were selectively etched out by immersing in buffered-oxide etchant for 9 min. After the local etching barrier was removed, the residual PR was removed with the help of acetone and IPA. With respect to the texturing process, samples were immersed in potassium hydroxide (KOH) at 80 °C and then in a distilled water solution of an additive for 30 min. Finally, any residual SiNx was removed by immersing the samples in diluted hydrogen fluoride for 5 min.

**Characterization.** Light absorbance, reflectance, and transmittance were measured by UV-vis spectroscopy (JASCO V-670 UV/Vis NIR spectrophotometer), and XRD (SmartLab, Rigaku) was performed using CuKα radiation (1.54 nm). JV scans were performed in the direction of the open-circuit voltage to the short-circuit current (i.e., reverse), and the voltage setting time was 200 ms. An XE-100 (Park Systems, Korea) AFM system was adopted for the c-AFM measurements. External quantum

efficiency measurements were conducted using a PV measurement system with a chopping frequency of 100 Hz.

The solar cell performance of perovskite solar cells were measured by a solar simulator (WACOM WXS-155S10 class AAA) with $100 \, mW \times cm^{-2}$ irradiation with Xe lamp. Modulated reference silicon solar cells were used for calibration prior to measurement. Photo-generation current as a function of voltage were measured by a source meter (Keithley 2400). All devices had measured using a proper size shadowing mask. The scan voltage setting time was 200 ms.

**μ-PL and μ-LBIC 3D mapping.** The μ-PL and μ-LBIC mappings were performed using a PL spectroscopy setup based on a confocal microscope. The samples were illuminated from the top. The point-shaped excitation and detection allows for diffraction-limited resolution while using an objective lens with a high numerical aperture (NA). In this case, an objective lens with an NA of 0.9 was used. For all measurements, an excitation wavelength of 635 nm, illumination intensity of 10 μW, and laser spot with a diameter of 0.5 μm were selected. For the detection of the PL signal, a silicon line CCD combined with a grating spectrometer was used, and this combination provided the PL spectrum for each pixel. To suppress the excitation light, a 700 nm longpass filter was applied. Both the spectral position and peak height were obtained by a Gaussian fit to the measurement data. The local light beam-induced current signal was amplified with the help of a low noise preamplifier.

**Reporting summary.** Further information on research design is available in the Nature Research Reporting Summary linked to this article.

## Data availability

The data that support the findings of this study are available within the Supplementary Information and from the corresponding author upon reasonable request.

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

## Acknowledgements

This research was supported by the Technology Development Program to Solve Climate Changes of the National Research Foundation (NRF) funded by the Ministry of Science and ICT (NRF-2017M1A2A2087351). This research was also supported by Korean Energy Technology Evaluation and Planning (KETEP) and the Ministry of Trade, Industry & Energy (MOTIE) of the Republic of Korea (no. 20173030014370).

## Author contributions

S.-W.L. contributed to the overall project. S.B., J.-K.H., W.L., S.L., J.Y.H., K.C., and S.K. fabricated the precursors and perovskite solar cells, and performed several measurements and analyses. F.D.H. and M.C.S. conducted μ-PL and μ-LBIC 3D mapping analysis. S.B.C. and W.M.K. conducted ITO layer fabrication analysis. D.C., D.K., J.Y., and S.J.P. performed patterned silicon substrate fabrication and analysis. S.J. performed stress simulation and analysis. S.G., Y.K., H.-S.L., and D.K. directed the project. All authors reviewed the manuscript.

## Competing interests

The authors declare no competing interests.
