## [Peer Review File · Communications Chemistry]

Reviewers' comments:

Reviewer #1 (Remarks to the Author):

In this paper, Lee et al. described the approach to place conformal perovskite on textured surfaces and investigating the effects on the perovskite layers. And they also analyzed perovskite on textured surfaces caused by the randomness of conventional silicon textures. It is a good addition to the reported perovskite/Si tandem solar cell and solar cell field. This paper could be published after they have addressed the following concerns.

- 1 The paper showed that "The current density exhibits the greatest correlation with the PCE. Statistically, if we can obtain 1 mA/cm² more current density, we will obtain 1.54 percentage point more efficiency" How they calculate this?
- 2 The TiO₂ they sputtered should have big effect on the tandem device. Why they choose this sputtering method? Recent a paper reported the TiO₂ matters. Sci. Adv. 2018;4:eaau9711
- 3 They authors could fabricate the same device but on normal planar Si as the control?
- 4 They found there excessive PbO left after annealing, is it possible that less PbO would be preferable or the annealing temperature affects?
- 5 The authors should have adopted different kind size Si texture substrate (like different patterning)?
- 6 Is it possible to get the average lifetime of electrons through PL measurement?
- 7 The cross-section TEM or SEM should be provide to further characterized the elements distribution.
- 8 From the figure 3, both samples showed the same the UV-visible absorption between 300-600nm but just different from 600-800nm?

Reviewer #2 (Remarks to the Author):

Recommendation: Reject

The manuscript by Lee et al. presents an interesting solution of depositing perovskite conformally on top of textured surfaces, such as pyramids on silicon. Such an approach is essential when making highly efficient perovskite/silicon tandems and would thus be of great interest to a broad range of readers. They support their findings with some analysis, however, I feel that in this form the publishing of the manuscript would be premature.

The following things should be considered when improving the manuscript:

1. The procedure of sputtering PbO and then converting it in dry process is relatively rare. The authors cite 3 similar reports with PbO [ref 34-36] but do not explain what the difference to their process is.
2. It would be very interesting to show more results and analysis of the process and films, such as temperature dependence, MAI amount, can you tune the PbI₂ excess with those parameters, some optimization processes
3. The lead excess might be due to non converted PbI₂ at the bottom of the layer, close to the surface. Is the PbO fully converted? Depth profiling, if possible, would be interesting. Additionally, by adding HF the authors change the porosity of the PbO. Does this change the lead excess? Only the perovskite peak in XRD is analysed.
4. The typical bandgap of MAPbI₃ is around 800 nm, however, there were some reports that evaporated perovskite (also a dry process) has a higher bandgap due to having a cubic structure. The bandgap reported here is 770 nm. What is the crystal structure of MAPbI₃ made with your process?
5. Authors show nice PL mapping images. A PL spectra is missing though.
6. Similarly, LBIC images are interesting but open some questions. What does a point with 0 PL and/or 0 LBIC current mean? A shunt?
7. Finally, any kind of JVs are missing. Authors were able to measure LBIC so some sort of JVs should be obtainable. Since the deposition of spiro or other HTL is not trivial (probably the reason

why authors deposited MoO₃ directly on perovskite), the JVs of device on flat glass substrates would already give some valuable information.

There are also some minor things to improve:

1. The authors make a good work of summarizing the recent results on perovskite/silicon tandems in the SI. However, a stronger motivation in the introduction is needed.
 - a. Not only perovskite, but also contact layers have to be deposited conformally on textures
 - b. There have been a few reports on simulations of perovskite/silicon tandems with textures (doi: 10.1039/C8EE02469C, 10.1364/OE.24.0A1288) which show that texture is indeed needed and tell what the gain would be
 - c. The bypass of the silicon front texture could be by having a foil on top (doi: 10.1039/C8EE02469C, 10.1021/acseenergylett.8b01201)
 - d. There is also a triple junction report on textured silicon (doi 10.1021/acseenergylett.8b01165)
 - e. The PCE of Sahli et al [ref] is exactly 25.2% (and not above 25.0%) since it was certified.
2. Table S4: a formula is missing how to calculate weighted reflectance

Reviewer #3 (Remarks to the Author):

The authors report the method to fabricate tandem solar cells by forming MAPI perovskite layer on textured silicon using the dry process. The characteristic feature of this formation of the perovskite layer seems to be that the valley portion of the texture can be completely covered with the perovskite layer by using the two-step method using the MAI from the PbO₂ layer. Such the fabrication method is a necessary technology for the preparation of perovskite tandem solar cells. Therefore, it may be important as engineering. However, scientific progress has been unclear as the authors have not presented effective performance as solar cells. Therefore, I cannot support this manuscript been published in Communications Chemistry.

Detailed Response to Reviewers

Reviewer(s)' Comments to Author:

Reviewer #1 (Remarks to the Author):

Comments:

In this paper, Lee et al. described the approach to place conformal perovskite on textured surfaces and investigating the effects on the perovskite layers. And they also analyzed perovskite on textured surfaces caused by the randomness of conventional silicon textures. It is a good addition to the reported perovskite /Si tandem solar cell and solar cell field. This paper could be published after they have addressed the following concerns.

- Thank you for your kind consideration and comments on important points. The answers to each comment and how it is reflected in the paper are listed below.

1. The paper showed that “The current density exhibits the greatest correlation with the PCE. Statistically, if we can obtain 1 mA/cm² more current density, we will obtain 1.54 percentage point more efficiency” How they calculate this?

- To find relationship between solar cell parameters and power conversion efficiency from the scattered data points, we had tried to apply linear regression fitting. For this purpose we had used linear regression fitting function in Origin software.
- Related sentences also added to the caption of Figure 1.
- *“The slopes had been obtained by linear regression fitting.”*
- Also, the graph is updated by adding the latest perovskite / silicon tandem data.

2. The TiO₂ they sputtered should have big effect on the tandem device. Why they choose this sputtering method? Recent a paper reported the TiO₂ matters. Sci. Adv. 2018;4:eaau9711

- The reason we chose sputtering TiO₂ is, we need a conformal ETL when we fabricate perovskite solar cells on textured silicon in the future. The solution method makes it difficult to form conformal ETLs on silicon textures with micrometer sizes. We recently published a paper using sputtering TiO₂ that achieved up to 20% perovskite solar cell efficiency in the flat surface. This is a reason why we chose TiO₂ as an ETL in this manuscript. (Lee, Sang-Won, et al. "Sputtering of TiO₂ for High-Efficiency Perovskite and 23.1% Perovskite/Silicon 4-Terminal Tandem Solar Cells." ACS Applied Energy Materials 2.9 (2019): 6263-6268)
- Related sentences also added to 51th line of page 4 of the main manuscript. The sentence added to the text is as follows.
- *“Furthermore, not only perovskite absorber layer, but also the other components like electron transfer layer (ETL) and hole transfer layer (HTL) have been produced by solution process”*

3. They authors could fabricate the same device but on normal planar Si as the control?

- As you recommended, we had tried to produce perovskite solar cell on flat surface. In order to minimize the influence of the substrate, the perovskite solar cells was fabricated on sputtered TiO₂/ FTO substrate. In our previous reports, we had achieved 20% by 1-step spin coating process with this substrate. (Lee, Sang-Won, et al. "Sputtering of TiO₂ for High-Efficiency Perovskite and 23.1% Perovskite/Silicon 4-Terminal Tandem Solar Cells." ACS Applied Energy Materials 2.9 (2019): 6263-6268)

→ Related sentences also added to 107th line of page 7 of the main manuscript. The sentence added to the text and figures are as follows.

→ “Based on the experiments about conversion conditions, perovskite solar cells were produced. Fig. 4 shows the solar cell parameter distributions produced at different conversion condition. Amount of MAI was fixed at 5.0 g. A 11.1% PCE was obtained at 200 °C 70 min conversion condition. For the perovskite solar cell, we used TiO₂ ETL and Spiro-OMeTAD HTL, which achieved around 20% in our previous paper.⁴⁰ The LIV curves and the maximum power point tacking results for each solar cells are in the Supplementary Fig. 8,9 and Supplementary Table 3.4.”

Fig. 4. Solar cell parameter distributions converted at different conditions. Amount of MAI was fixed at 5.0 g. Eight solar cells were produced for each condition. **a)** Open-circuit voltage, **b)** short-circuit current density, **c)** fill factor, and **d)** power conversion efficiency. Hysteresis was large when measuring solar cell efficiency, and there were cases where FF was over estimated. Each data point was adopted from the LIV reverse scan results and the measurements were conducted with 0.075 cm² shadow mask under the AM1.5G condition.

Supplementary Fig. 8. LIV measurement result of PbO converted perovskite solar cells produced at different conversion conditions.

Supplementary Table 3. Solar cell parameters with different conversion conditions.

Reverse	V_{oc} (V)	J_{sc} (mA/cm²)	Fill Factor (%)	R_{sh} (ohms)	R_s (ohms)	PCE (%)
150 °C 210 min	0.642	11.4	72.6	-800	178	5.3
170 °C 210 min	0.714	12.6	87.1	-503	138	7.9
200 °C 15 min	0.708	6.7	71.8	-1570	401	3.4
200 °C 35 min	0.652	20.4	53.5	-2460	122	7.1
200 °C 70 min	0.791	19.5	71.2	-699	79.8	11.1
200 °C 140 min	0.735	21.4	60.3	-3000	82.8	9.5
200 °C 210 min	0.649	22.4	45.1	6450	157	6.6
Forward						
150 °C 210 min	0.399	13.4	14.3	139	610	0.8
170 °C 210 min	0.394	11.3	12.7	122	937	0.6
200 °C 15 min	0.402	6.3	12.5	187	1870	0.3
200 °C 35 min	0.388	15.9	14.8	154	699	0.9
200 °C 70 min	0.496	22.9	15.2	330	621	1.7
200 °C 140 min	0.530	21.5	21.2	300	283	2.4
200 °C 210 min	0.576	22.2	23.9	573	381	3.1

Supplementary Fig. 9. The 500 seconds maximum power point tracking result of PbO converted perovskite solar cells produced at different conversion conditions.

Supplementary Table. 4. Current density before and after 500 seconds maximum power point tracking under the AM1.5G condition.

	Pristine current density (mA/cm ²)	500 seconds after current density (mA/cm ²)
150 °C 210 min	11.4	9.3
170 °C 210 min	12.7	13.0
200 °C 15 min	6.7	3.2
200 °C 35 min	18.4	12.0
200 °C 70 min	19.5	17.2
200 °C 140 min	21.4	14.0
200 °C 210 min	21.0	15.2

4. They found there excessive PbO left after annealing, is it possible that less PbO would be preferable or the annealing temperature affects?

- ➔ As you recommended, to demonstrate the effect of process condition on remaining PbO, we had conducted conversion experiment with different time, temperature, and amount of MAI. The original Figure 3 is replaced with experimental results.
- ➔ The second reviewer asked the same question. In detail response and data to this question, we would be grateful if you could refer to our answers to question 2 of the second reviewer below.

5. The authors should have adopted different kind size Si texture substrate (like different patterning)?

- As you recommended, we had conducted PL mapping using different sized v-groove textures. The PL intensity and the peak position difference tendency were the same even when the length of the slope was changed.
- Related sentences were added to Fig. 7 caption in page 12 of the main manuscript. The sentence and figures added to the text are as follows.
- “The results of PL mapping using different size V-groove textures are in Supplementary Fig. 13. The PL intensity and the peak position tendency were the same even when the length of the slope was changed.”

Supplementary Fig. 13. Combined image of the collective 2D maps for v-groove texture and v-groove texture with longer slope. **a),b)** Focus height, **c),d)** PL peak position, **e),f)** normalized PL intensity, and **g),h)** corresponding SEM image.

6. Is it possible to get the average lifetime of electrons through PL measurement?

- We measured time resolved-PL (TRPL) with PbO converted perovskite on V-groove. The structure of the sample in the measurement was Perovskite / $c\text{-TiO}_2$ / V-groove Si. In the Pristine state, a life time of less than 20 ns was obtained for the top, valley, and slope. When using HF treated PbO, the life time was improved to about 70 ns. However, further experimentation is still needed to determine whether this is actually a result of the improved film quality or not. This is because HF can also etch TiO_2 layer, which can act as carrier quenching layer. Therefore, attaching life time data to the main manuscript at the current level can be controversial.

7. The cross-section TEM or SEM should be provide to further characterized the elements distribution.

- We have tried to measure the depth profile with TEM and SIMS (Secondary Ion Mass Spectrometry). However, we could obtain contradictory profile between TEM EDS line profile and SIMS data. It may be confusing to publish the data obtained so far. So we think we'd better not use this data in the main manuscript. To solve this problem, the specimens made under various conditions have to be tested with various measurements. We can further analyze the depth profile by of our sample by atom probe tomography (APT), XPS depth profile, and auger electron spectroscopy.

8. From the figure3, both samples showed the same the UV-visible absorption between 300-600nm but just different from 600-800nm?

- This may be due to the difference in thickness, morphology, quantity of perovskite layer, and quality of the perovskite film. To demonstrate this problem, we replaced Figure. 3 with a more informative figure. Looking at Figure 3, it can be seen that the XRD data and the absorbance data have similar tendency. These results suggest that an important part of the difference in light absorbance comes from the difference in the quantity of perovskite converted inside the thin film.

Reviewer: 2

The manuscript by Lee et al. presents an interesting solution of depositing perovskite conformally on top of textured surfaces, such as pyramids on silicon. Such an approach is essential when making highly efficient perovskite/silicon tandems and would thus be of great interest to a broad range of readers. They support their findings with some analysis, however, I feel that in this form the publishing of the manuscript would be premature.

The following things should be considered when improving the manuscript:

- Thank you for your detail practical review and comments on important points. The answers to each comments and how it is reflected in the paper are listed below.

1. The procedure of sputtering PbO and then converting it in dry process is relatively rare. The authors cite 3 similar reports with PbO [ref 34-36] but do not explain what the difference to their process is.

- Thank you for your comment. To demonstrate the differences between previously reported results and our paper, we have added explanation previous reports.
- Related sentences were added to 80th line of page 5 of the main manuscript. The sentence added to the text is as follows.
- "We had adopted the PbO precursor which can be easily deposited by dry process. Perovskite solar cells with PbO based conversion process previously reported on flat surface.^{36, 37, 38} Zhang, Z. et al had reported 14.1% with sputtered PbO and MAI solution dipping conversion process.³⁶ Yan-Lin Song et al had reported 14.6% with PbO electro deposition and MAI spin coating conversion.^{37, 39}
- In this case, with the textured surface and a solution-based process, conformal films were not obtained even with a conformal precursor (Supplementary Fig. 1)."

2. It would be very interesting to show more results and analysis of the process and films, such as temperature dependence, MAI amount, can you tune the Pbl₂ excess with those parameters, some optimization processes

- As you recommended, for the demonstration of process parameter dependence of perovskite films produced by PbO dry conversion process, we had conducted conversion experiment with different time, temperature, and amount of MAI. The original Figure 3 is replaced with experimental results.
- Related sentences were added to 92th line of page 7 of the main manuscript. The sentence and figures added to the text are as follows.
- "Conformal 80 nm PbO films were sputtered and converted into CH₃NH₃Pbl₃ layers through a direct contact reaction with CH₃NH₃I powder at 100, 150, 170, 200, 250 °C for 35, 70, 140, 210 min. As shown in Fig 3a, as reaction temperature and time increased, the films gradually changed to yellow Pbl₂ and then turned into a dark brown colored perovskite. In the case of 250 °C, MAI was burned and stuck to the substrate. In the process of removing MAI, all the films were removed together. These tendency was also observed at light absorptance at 550 nm, SEM image, and normalized XRD results as shown in Fig. 3b, c and d. Normalization was conducted based on the maximum intensity to the ratio of PbO_x, Pbl₂, CH₃NH₃Pbl₃ intensity. The peak positions of each component were 30.5°, 12.8°, and 14.3° degrees.³⁷ The cross section of the SEM, the light absorption at the full wavelength, the diffraction intensity of the XRD at the entire angle, shorter time conversion result at 200 °C, 250 °C, tauc plot result for optical bandgap calculation, and the effect of the MAI amount on the conversion process is shown in Supplementary Fig. 2-7. The dependence on the amount of MAI in the reaction was not significant within the experiment conditions.

Fig. 3. Characteristic of films produced by the dry two-step conversion on a flat surface with different conversion time and temperature. **a)** Digital camera image of corresponding films on a 1.5 cm X 1.5 cm TiO₂/FTO substrate, **b)** light absorbance at 550 nm, and **c)** scanning electron microscopy (SEM) top view. **d)** Normalized XRD peak intensity at 30.5° for PbO, 12.8° for PbI₂ and 14.3° for CH₃NH₃PbI₃.

Supplementary Fig. 2. Light absorptance from 300 nm to 900 nm of films which produced by corresponding conversion temperature and time

Supplementary Fig. 3. SEM cross section image of produced films depend on process temperature and time.

Supplementary Fig. 4. XRD peak intensity of films which produced by corresponding conversion temperature and time. **a)** 100 °C, **b)** 150 °C, **c)** 170 °C, **d)** 200 °C, and **e)** 250 °C

Supplementary Fig. 5. Image of films which fabricated with 2, 5, and 15 minute conversions at 200 and 250 °C. **a)** Digital camera image, **b)** SEM top view, **c)** SEM cross section view.

Supplementary Fig. 7. Conversion process results of 70 minute conversion process at 200 °C depending on the amount of MAI. a) Digital camera image of different amount of MAI on PbO / TiO₂ / FTO film. b) perovskite film digital camera image after conversion process, c) light absorbance, d) XRD peaks, and e) SEM images.

3. The lead excess might be due to non converted PbI_2 at the bottom of the layer, close to the surface. Is the PbO fully converted? Depth profiling, if possible, would be interesting. Additionally, by adding HF the authors change the porosity of the PbO . Does this change the lead excess? Only the perovskite peak in XRD is analysed.

- We have tried to measure the depth profile with TEM and SIMS (Secondary Ion Mass Spectrometry). However, we could obtain contradictory profile between TEM EDS line profile and SIMS data. It may be confusing to publish the data obtained so far. So we think we'd better not use this data in the main manuscript. To solve this problem, the specimens made under various conditions have to be tested with various measurements. We can further analyze the depth profile by of our sample by atom probe tomography (APT), XPS depth profile, and auger electron spectroscopy.
- The XRD of the total angle was compared with the perovskite prepared by the 1-step solution method, perovskite prepared with PbO , and HF-treated PbO . Under current experimental conditions, HF treatment does not appear to affect the reduction of excess Pb. Rather, the reaction temperature and time seem to be more important, as we have shown in manuscript Fig. 3 at page 6 and in supporting information Supplementary Fig. 21 at page 29.

Supplementary Fig. 21. The XRD of the total angle was compared with the perovskite prepared by the 1-step solution method, perovskite prepared with PbO , and HF-treated PbO .

4. The typical bandgap of MAPbI_3 is around 800 nm, however, there were some reports that evaporated perovskite (also a dry process) has a higher bandgap due to having a cubic structure. The bandgap reported here is 770 nm. What is the crystal structure of MAPbI_3 made with your process?

- Thank you for the important comment. In order to find the bandgap of the produced perovskite, we applied a Tauc plot. As a result, our perovskite has a band gap of about 1.60 eV. However, this may not be enough to confirm the crystal structure. Because our perovskite surface was rather rough and there is variation in thickness. As a result, our perovskite may not at the optimal condition for Tauc plot application.
- Related sentences and figures were added to 101th line of page 7 of the main manuscript.
- *"The cross section of the SEM, the light absorption at the full wavelength, the diffraction*

intensity of the XRD at the entire angle, shorter time conversion result at 200 °C, 250 °C, tauc plot result for optical bandgap calculation, and the effect of the MAI amount on the conversion process is shown in Supplementary Fig. 2-7.”

Supplementary Fig. 6. Tauc plot results of perovskite thin films fabricated at different temperatures and reaction times. The optical band gap was calculated between 1.60eV and 1.62eV.

5. Authors show nice PL mapping images. A PL spectra is missing though.

- ➔ As you recommended, we had constructed PL spectrum using corrective 2D PL image at focus top, focus middle and focus bottom. PL spectrum at denoted points in the below image were used.
- ➔ Related sentence and figure were added to Fig.7 caption in page 12 of the main manuscript. The sentence added to the text is as follows.
- ➔ “The results of displaying the PL spectrum at a specific position for specific focus heights are shown in the Supplementary Fig. 14.”

Supplementary Fig. 14. The results of displaying the PL spectrum at a specific position for specific focus heights.

6. Similarly, LBIC images are interesting but open some questions. What does a point with 0 PL and/or 0 LBIC current mean? A shunt?

- ➔ Thank you for your comment. There can exist two possibilities. The first is when the points are not actually zero and the next is when the points are not really zero. To verify those cases, we investigated the raw data.
- ➔ 1. Looking at the raw data, most of the dark areas of the PL mapping and LBIC data actually showed very small values rather than actual zero.
- ➔ 2. But there were actually zero part of the raw data. These parts can act as shunt points. This kind of problem can be occurred during the deposition of HTL, butter layers, and transparent electrodes sputtering process.
- ➔ Related sentences were added to Fig. 8 caption in page 13 of the main manuscript. The sentence and figures added to the text is as follows.
- ➔ “Most of the dark areas of the PL and LBIC mapping actually showed very small values rather than actual zero. But there were actually zero part in the raw data. These parts can be shunt points. This kind of problem can be occurred during the deposition of HTL, butter layers, and transparent electrodes sputtering process.”

7. Finally, any kind of JVs are missing. Authors were able to measure LBIC so some sort of JVs should be obtainable. Since the deposition of spiro or other HTL is not trivial (probably the reason why authors deposited MoO3 directly on perovskite), the JVs of device on flat glass substrates would already give some valuable information.

- As you recommended, we had tried to produce perovskite solar cell on flat surface. In order to minimize the influence of the substrate, the perovskite solar cells was fabricated on sputtered TiO₂/ FTO substrate. In our previous reports, we had achieved 20% by 1-step spin coating process with this substrate. (Lee, Sang-Won, et al. "Sputtering of TiO₂ for High-Efficiency Perovskite and 23.1% Perovskite/Silicon 4-Terminal Tandem Solar Cells." ACS Applied Energy Materials 2.9 (2019): 6263-6268)
- The first reviewer asked the same question. In detail response and data to this question, we would be grateful if you could refer to our answers to question 3 of the first reviewer above.

8. There are also some minor things to improve:

8.1 The authors make a good work of summarizing the recent results on perovskite/silicon tandems in the SI. However, a stronger motivation in the introduction is needed.

- Thank you for your important and essential comment. To strengthen the motivation, we updated our perovskite/silicon tandem review table with latest reports and revised the introduction of the main manuscript based on your comments below.

a. Not only perovskite, but also contact layers have to be deposited conformally on textures

- To deal with this problem, we have added explanation about fabricating conformal charge transfer layers and the other components.
- Related sentences also added to 51th line of page 4 of the main manuscript. The sentence added to the text is as follows.
- "Furthermore, not only perovskite absorber layer, but also the other components like electron transfer layer (ETL) and hole transfer layer (HTL) have been produced by solution process."

b. There have been a few reports on simulations of perovskite/silicon tandems with textures (doi: 10.1039/C8EE02469C, 10.1364/OE.24.0A1288) which show that texture is indeed needed and tell what the gain would be

c. The bypass of the silicon front texture could be by having a foil on top (doi: 10.1039/C8EE02469C, 10.1021/acsenergylett.8b01201)

- As you recommended, we have added the explanation about reports on simulations of perovskite/silicon tandems with textures and adopting anti reflection foils on top of the tandem device. (doi: 10.1039/C8EE02469C, 10.1364/OE.24.0A1288, 10.1021/acsphotonics.7b00138)
- Related sentences also added to 58th line of page 4 of the main manuscript. The sentence added to the text is as follows.
 - "The use of antireflection foils on top of the tandem device has been studied as one way to achieve high current density. Referring to literature results, however, eventually forming perovskite on textured silicon can achieve maximum current density."^{24,25,26}

24. Jost M, et al. Textured interfaces in monolithic perovskite/silicon tandem solar cells: advanced light management for improved efficiency and energy yield. Energy & Environmental Science 11, 3511-3523 (2018).

- 25. Santbergen R, et al. Minimizing optical losses in monolithic perovskite/c-Si tandem solar cells with a flat top cell. *Opt Express* **24**, A1288-A1299 (2016).
- 26. Jost M, et al. Efficient Light Management by Textured Nanoimprinted Layers for Perovskite Solar Cells. *Acs Photonics* **4**, 1232-1239 (2017).

d. There is also a triple junction report on textured silicon (doi 10.1021/acsenergylett.8b01165)

- As you told us, We've updated the Perovskite/Silicon Tandem review table with your feedback and the latest results.
- Related table was added to Supplementary Table 1 of the Supplementary Information

e. The PCE of Sahli et al [ref] is exactly 25.2% (and not above 25.0%) since it was certified.

- As you mentioned, we changed the 'above 25%' notation to '25.2%' at 42th line of page 3.

8.2. Table S4: a formula is missing how to calculate weighted reflectance

- To deal with this point, we added equation for solar weighted reflectance (SWR) at supplementary equation 1 of the Supplementary Information. The sentence added to the text is as follows.
- "Solar weighted reflectance (SWR) was calculated based on the supplementary equation

1 56."

$$\text{Solar Weighted Reflectance (SWR)} = \frac{\int_{300 \text{ nm}}^{1200 \text{ nm}} R(\lambda) N_{\text{photon}}(\lambda) d\lambda}{\int_{300 \text{ nm}}^{1200 \text{ nm}} N_{\text{photon}}(\lambda) d\lambda}$$

$R(\lambda)$: spectral reflectivity,

N_{photon} : photon number of the solar irradiation (AM1.5G) per unit area per unit wavelength.

Supplementary Equation 1. Equation for solar weighted reflectance calculation."

56. Sai H, Kanamori Y, Arafune K, Ohshita Y, Yamaguchi M. Light trapping effect of submicron surface textures in crystalline Si solar cells. *Prog Photovoltaics* **15**, 415-423 (2007).

Reviewer #3 (Remarks to the Author):

Comments:

The authors report the method to fabricate tandem solar cells by forming MAPI perovskite layer on textured silicon using the dry process. The characteristic feature of this formation of the perovskite layer seems to be that the valley portion of the texture can be completely covered with the perovskite layer by using the two-step method using the MAI from the PbO₂ layer. Such the fabrication method is a necessary technology for the preparation of perovskite tandem solar cells. Therefore, it may be important as engineering. However, scientific progress has been unclear as the authors have not presented effective performance as solar cells. Therefore, I cannot support this manuscript been published in Communications Chemistry.

- Thank you for your important comments.
- For the scientific progress, the main demonstration of this paper is to present novel fabrication method for the formation of several hundred nanometers of perovskite layer on several micrometer silicon texture. Also, we had tried to produced characterization method with patterned v-groove texture by using high resolution PL and LBIC 3D mappings. These methods can eliminate problems originated by randomness of silicon texture. Also, it can produce comprehensive information about the perovskite on textured silicon surface.
- Furthermore, as you mentioned, we had tried to demonstrate the performance of solar cells fabricated by PbO dry conversion process. We added below figures and table to the main manuscript and supplementary information.
- **Fig. 1.** Characteristic of films produced by the dry two-step conversion on a flat surface with different conversion time and temperature.
- **Fig. 4.** Solar cell parameter distributions converted at different conditions.
- **Supplementary Fig. 8.** LIV curve of PbO converted perovskite solar cells produced at different conversion conditions.
- **Supplementary Table. 3.** Solar cell parameters with different conversion conditions.
- **Supplementary Fig. 9.** The 500 seconds maximum power point tracking result of PbO converted perovskite solar cells produced at different conversion conditions.
- **Supplementary Table. 4.** Current density before and after 500 seconds maximum power point tracking under the AM1.5G condition.
- The first and second reviewers asked the same question. In detail response and data to this question, we would be grateful if you could refer to our answers to question 3 of the first reviewer and the question 2 of the second reviewer.

Reviewers' comments:

Reviewer #1 (Remarks to the Author):

The authors have fully addressed my concerns. After the revision, it is greatly improved and I think it could be published in this journal.

Reviewer #2 (Remarks to the Author):

The authors have put a lot of effort into optimizing their procedure and consequently significantly improved the experimental part of the manuscript. However, the discussion of the new results is missing and is required before the manuscript can be published. A lot of new, interesting data were added but their meaning could be strengthened.

Additionally, some minor questions also remain.

- Line 52 in the main text: When depositing layers on textured surfaces, they have to be grown conformally, which is very hard by solution processing. To have a working cell, conformal deposition of charge transport layers is critical, in the same way as the authors are investigating conformal deposition of perovskite!
- Caption of the figure 8. Buffer, not butter.
- Figure 10, instead of figure 4 at the end of the text (page 16).
- Line 83: "In our case" would make the text much clearer
- In supplementary figure 7, what have they done with excess (unreacted) MAI? Removed by solvent (e.g. isopropanol) or mechanically removed (nitrogen gun)?
- In supplementary figure 7, labelling FTO and perovskite would help distinguish between the two layers and help estimate the layer thickness
- The authors should comment about possible reasons for the JV behavior in Supplementary Figure 8.
- In supplementary figure 9 they state they measured MPP, but show current tracking.
- Supplementary figure 14: Does this mean that the focus position is critical when determining the PL? Should the scans be done for different focuses since by changing the focus the intensity changes? Could the authors also add a figures where they overlay PL spectra for points 2, 4 and 5 for different focuses, respectively?

After this corrections the manuscript should be ready for publishing.

Detailed Response to Reviewers

Reviewers' comments:

Reviewer #1 (Remarks to the Author):

Comments:

The authors have fully addressed my concerns. After the revision, it is greatly improved and I think it could be published in this journal.

- Thank you very much for your kind comments. With your comments, this paper improved very much. It was an honor to receive comments from you.

Reviewer #2 (Remarks to the Author):

Comments:

The authors have put a lot of effort into optimizing their procedure and consequently significantly improved the experimental part of the manuscript. However, the discussion of the new results is missing and is required before the manuscript can be published. A lot of new, interesting data were added but their meaning could be strengthened. Additionally, some minor questions also remain.

- Thank you very much for your detail and thoughtful comments. With your precise and important comments, this paper improved very much. It was an honor to receive a comment from you, and it was a great opportunity to develop this paper.
- The main manuscript and supplementary information have been revised again to strengthen the meaning of our finding and to express more concise way. Revised parts are highlighted in the manuscripts.
- Detail response for your respective comments have written below.

1. Line 52 in the main text: When depositing layers on textured surfaces, they have to be grown conformally, which is very hard by solution processing. To have a working cell, conformal deposition of charge transport layers is critical, in the same way as the authors are investigating conformal deposition of perovskite!

- Yes you are right. This is also very important issues in producing perovskite solar cells in textured silicon solar cells. To address charge transport layer issues, we have added below sentence to the main manuscript 51th line and 63th line in page 4.
- "However, most perovskite solar cells are manufactured by solution-based processes that cannot be used on the micrometer-sized textured structure. Not only perovskite absorber layer, but also the other components like electron transfer layer (ETL) and hole transfer layer (HTL) have been mainly produced by solution process."
- "Therefore, a technique to produce a conformal perovskite layer and other solar cell components on a textured surface is required."

2. Caption of the figure 8. Buffer, not butter.

3. Figure 10, instead of figure 4 at the end of the text (page 16).

- 'butter' to 'buffer' at Figure 8 caption, page 13.
- 'Fig. 4' to 'Fig. 10', page 17.

4. Line 83: "In our case" would make the text much clearer

- We have added 'In our case', in 83th line, page 5,

5. In supplementary figure 7, what have they done with excess (unreacted) MAI? Removed by solvent (e.g. isopropanol) or mechanically removed (nitrogen gun)?

- We had removed the excess MAI firstly with nitrogen gun and IPA spin drying. Detail removal process description is added to line 97, page 7 and line 268, page 19.

- “After the reaction finished, In the process of removing reminded MAI by nitrogen gas (N₂) blowing and isopropyl alcohol (IPA) rinsing, all the films were removed together.”
- “The MAI powder was directly spread over the precursor film and annealed at respective temperature for respective times in air. After the reaction finished, reminded MAI were removed by N₂ gas blowing, IPA rinsing and dried by spinning.”

6. In supplementary figure 7, labelling FTO and perovskite would help distinguish between the two layers and help estimate the layer thickness

- We have added inset color in Supplementary Fig 7e to help distinguishing between layers.

7. The authors should comment about possible reasons for the JV behavior in Supplementary Figure 8.

- According to the reference, *Snaith, H. J, et al. Anomalous hysteresis in perovskite solar cells. The journal of physical chemistry letters 5(9), 1511-1515 (2014).* The main reasons for JV hysteresis are known as 1) charge transfer rate difference between electron transfer layer and hole transfer layer, 2) ion migration in perovskite layer. The JV hysteresis in our case, is supposed to be caused by excessive ions, which were generated during the solid to solid conversion process. However, the exact reason have to be investigated in the continuous work. By solving the hysteresis problem, the device efficiency can be enhanced more.
- Related sentence and references are added to Supplementary Fig. 8 caption, page 11.
- “A hysteresis behavior can be originated from excessive ions, existing as interstitial defects, generated during dry two-step solid to solid conversion process⁵⁶. In continuous work, the reason for the hysteresis behavior in our perovskite films should be investigated further.”
- 56. Snaith, H. J, et al. Anomalous hysteresis in perovskite solar cells. The journal of physical chemistry letters 5(9), 1511-1515 (2014).

8. In supplementary figure 9 they state they measured MPP, but show current tracking.

- Yes you are right. It was our mistake. We have revised ‘MPP’ to ‘current tracking’ and related sentences in main manuscript and supplementary information.
- Main manuscript Page 8, 116th line. “The LIV curves and the current tracking results for each solar cells are shown in the Supplementary Fig. 8,9 and Supplementary Table 3,4.”
- Supplementary Fig. 9 caption, Page 13, “The 500 seconds current tracking result of PbO converted perovskite solar cells produced at different conversion conditions.”
- Supplementary Table. 4 caption, Page 13, “Current density before and after 500 seconds

current tracking under the AM1.5G condition.”

9. Supplementary figure 14: Does this mean that the focus position is critical when determining the PL? Should the scans be done for different focuses since by changing the focus the intensity changes? Could the authors also add a figures where they overlay PL spectra for points 2, 4 and 5 for different focuses, respectively?

After this corrections the manuscript should be ready for publishing.

- Data shown in Supplementary Fig 14 shows why 3D scanning is important when analyze conformal perovskite film on textured silicon surface. To obtain high resolution information, a 2D scan cannot provide accurate data because of the strong focus height dependence. To analyze perovskite at specific position on textured surface with high XYZ resolutions, 2D PL intensity and peak position have to be collected throughout whole silicon texture thickness at different focus height and all the data should be combined into a comprehensive single 2D image. Fig. 7 in the main manuscript summarize these process and shows combined images.
- Related sentences have added to the main manuscript line 152, page 11 and Supplementary Fig. 14 caption, in page 19.
- Since several hundred nanometers of perovskite layer is located on the micrometer-sized pyramidal textured silicon surface, and the measurements have to maintain high spatial and depth resolution simultaneously, only XY 2D mapping cannot provide accurate information, as shown Supplementary Fig. 14.
- It shows why 3D scanning is important when analyze conformal perovskite film on textured silicon surface. To analyze perovskite at specific position on textured surface with high XYZ resolutions, 2D PL intensity and peak position have to be collected throughout whole silicon texture thickness at different focus height and all the data should be combined into a comprehensive single 2D image. Fig. 7 in the main manuscript summarize these process and shows combined images

REVIEWERS' COMMENTS:

Reviewer #2 (Remarks to the Author):

The authors have improved the manuscript according to the concerns raised.